# A lightweight intelligent compression method for fast Sea Level Anomaly data transmission

**Xiaodong Ma**, **Xiang Wan, Lei Zhang***, **Dong Wang, Zeyuan Dai**

Department of Military and Marine Surveying and Mapping, Dalian Naval Academy, Dalian, China

* stone333@tom.com

## Abstract

Traditional compression methods struggle to preserve critical mesoscale ocean features like vortices during bandwidth-constrained marine data transmission. To aaddress this limitation, we propose CompressGAN, a novel deep learning framework that transcends conventional approaches reliant on generic image metrics (e.g., peak signal-to-noise ratio, PSNR; structural similarity index, SSIM). The architecture integrates global-local dual discriminators to enforce spatiotemporal coherence of mesoscale vortices, employs dilated convolutions to enhance feature receptive fields without computational overhead, and incorporates vortex recognition rate as a physics-aware evaluation metric. Furthermore, parametric pruning and adaptive quantization strategies are embedded to optimize memory efficiency for shipborne hardware constraints. Validation across multiple ocean reanalysis datasets demonstrates CompressGAN's superiority at 4 × compression ratios, achieving 91.46% mesoscale eddy identification accuracy (Iden) versus SRGAN (89.71%) and SRRes-Net (89.82%), while maintaining operational efficiency (148 s/image inference time, 25 GB peak memory). Generalization tests reveal controlled performance degradation: PSNR reduced by 4.2 ± 0.3 dB, SSIM by 0.7126, and Iden by 4.1%, confirming robustness under marine operational scenarios. This work resolves the critical trade-off between vessel-mounted computational limits and real-time ocean data demands, providing a viable pathway for integrated shipboard systems to reconcile multimodal marine data processing with navigation service requirements.

## 1. Introduction

With the rapid development of ocean observation technology, grid ocean data with high spatial and temporal resolutions have become an important basis for modern marine scientific research [1–4]. The wide application of satellite remote sensing, numerical models, and distributed sensor networks has enabled the acquisition frequency and coverage of global marine environmental parameters to reach unprecedented levels. These cubes contain physical parameters such as temperature, salinity, and velocity (in this paper, the sea surface height anomaly is the main

**Data availability statement:** All SLA files are available from the AVISO database. AVISO data can be downloaded (open acess) from https://www.aviso.altimetry.fr/en/my-aviso-plus.html.

**Funding:** The author(s) received no specific funding for this work.

**Competing interests:** The authors have declared that no competing interests exist.

research direction). However, the contradiction between the exponential growth of the data volume and the demand for long-distance transmission is becoming increasingly prominent, especially in special scenarios such as deep sea observations and polar research; thus, achieving efficient and reliable data transmission has become a technical bottleneck restricting real-time marine monitoring and collaborative research.

Existing data compression techniques face significant challenges in dealing with grid ocean data. Although conventional lossless compression algorithms (such as GZIP [5] and LZ77 [6]) can maintain data integrity, it is usually difficult for their compression rates to exceed 50%, and the computational complexity increases nonlinearly as the data dimensions increase. Although lossy compression methods (such as the discrete cosine transform and wavelet transform) can improve the compression efficiency [7], they may destroy the unique spatial continuity and physical conservation characteristics of marine data. For example, overcompression of thermohaline profile data leads to spatial gradient distortion of mesoscale vortex structures, which affects the initial field accuracy of ocean circulation models [8]. The special data formats developed in recent years for earth science research (such as NetCDF and HDF5) have improved the storage efficiency through metadata optimization and dimension reorganization [9], but they have not fundamentally solved the bandwidth constraint problem in the transmission stage. The emerging intelligent compression technology provides new possibilities for data volume reduction [10–14]. Moreover, feature extraction methods based on deep learning, such as convolutional autoencoders, can identify nonlinear coupling relationships between ocean parameters and achieve high magnification compression through latent spatial mapping. Studies have shown that such methods can achieve a compression rate of more than 90% for data with strong spatial correlation, such as ocean and atmosphere parameterized grid data, while maintaining the main dynamic characteristics [15–19]. For Marine satellite remote sensing data, Fanelli et al., based on an expanded super-resolution network, achieved an effective resolution improvement for the publicly available abnormal sea surface temperature data and verified the validity of the data from a sub-mesoscale perspective [20]. Nardelli et al. achieved targeted resolution improvement for SLA data by adding an attention mechanism to the existing super-resolution network. They verified it using the MSE metric among multiple baseline models and obtained good model verification effects in the experimental results [21]. Cui et al. used an improved super-resolution model of deep learning gradient constraint embedding method to enhance the resolution of sea surface height data and obtained experimental results with more promising application prospects in Marine engineering [22].

However, data compression requirements and application accuracy in different scenarios are not completely uniform. In current relevant studies, data accuracy evaluation of data compression and decompression processes still utilize image processing values such as the structural similarity index (SSIM), mean squared error (MSE), and peak signal-to-noise ratio (PSNR) as the final evaluation index, rather than a specific parameter of concern to data users. Therefore, in this paper, we propose an intelligent compression framework for lightweight marine grid data for

long-distance transmission and ship-based use. By integrating physical prior knowledge and a data-driven model, a multi-scale adaptive quantization mechanism is constructed to effectively control the amount of transmitted data on the premise of ensuring the analyzability of dynamic ocean processes. This method pays special attention to the full-link consistency of the transmission-reconstruction-application of sea level anomaly (SLA) and adopts the forward validation strategy based on the ocean numerical model to ensure that the compressed data can meet the accuracy requirements in different application scenarios. The experimental results show that the comprehensive performance of the framework on typical marine datasets is better than those of the existing mainstream methods and provides a new technical path for building an efficient and reliable marine data transmission system.

## 2. Data and research area

### 2.1. Sea level anomaly data

The grid data (mean sea level anomaly, MSLA) were provided by the Archiving, Validation, and Interpretation of Satellite Oceanographic (AVISO) program in France. The dataset combines altimetry data from multiple satellites such as Topex/Poseidon (T/P), Jason-1, European Remote Sensing Satellite (ERS), and Environmental Satellite (ENVISAT). The temporal resolution of the data is 7 days, the spatial resolution is 1° × 1°, and the time span is January 1993 to July 2021. Necessary standard corrections have been made to the data, such as ionospheric delay correction, tropospheric dry and wet component correction, earth and sea tide correction, ocean load tidal correction, polar tide correction, electromagnetic deviation correction, instrument correction, and inverse barometer correction [23–25].

### 2.2. Ocean reanalysis data

The Japan Coastal Ocean Predictability Experiment 2 Modified (JCOPE2M) is a high-resolution reanalysis dataset for the Northwest Pacific Ocean developed by the Japan Marine Affairs Agency and has a horizontal grid resolution of 1/12° (about 8 km). The vertical full depth is divided into 46 layers, and the temporal resolution is updated daily [26,27]. By assimilating sea surface temperature (SST), sea surface height anomaly (SSHA), and some Argo buoy observations, the dataset achieves accurate reconstruction of the three-dimensional thermohaline structure and flow field of mesoscale vortexes. Previous studies have shown that JCOPE2M has a high reliability in the study of circulation dynamics in the Northwest Pacific Ocean, and it has been widely used in research on the variation mechanism of the Kuroshio extension body and vortex-current interaction.

The hybrid coordinate ocean model (HYCOM) is a numerical model of the global ocean developed by the U.S. Naval Research Laboratory (NRL) that uses a hybrid vertical coordinate system to optimize the simulation of ocean processes at different depths. The base version provides 1/12° (about 9 km) global coverage and a 1/25° (about 4 km) regional enhanced resolution, as well as a time resolution of hourly output and support for operational forecasting of up to 7 days [28–30]. The system can output multi-factor forecast products such as the sea surface height, two-dimensional positive pressure flow field, three-dimensional flow field, temperature, and salinity, and it can integrate historical observation data through reanalysis technology to provide multi-spatiotemporal scale data support for the study of dynamic ocean processes.

The China Ocean Reanalysis and Prediction System (CORA) is a high-precision regional ocean reanalysis dataset developed by the National Oceanic Information Center of China. It focuses on the Northwest Pacific Ocean and the offshore waters of China. This dataset integrates multi-source data such as satellite remote sensing data and buoy observations and adopts advanced data assimilation technology to couple regional ocean models to generate daily three-dimensional marine environment fields from 2000 to 2022. It has a horizontal resolution of 1/12° (about 9 km) and 40 vertical layers and includes core parameters such as the SST, salinity, and ocean currents. Verification has shown that the error of its sea surface temperature is less than 0.5°C, and it has been widely used in the study of ocean dynamic processes and operational forecasting [31].

## 2.3. Research area

In this study, we focused on the northwest Pacific region (0–60°N, 100–180°E) as the main research area. This area has two high-risk areas that contain the mesoscale vortex [32–35], the Kuroshio extension body area (KE, 30–45°N, 130–180°E), and the zone of the subtropical countercurrent (STCC, 15–28°N, 120–170°E)

## 3. Methods

### 3.1. Establishing the sample dataset

In the main research content of this study, data compression was realized using the following super-resolution enhancement process after sampling, so the dataset needed to provide a set of high-resolution images and a set of low-resolution image sample data. To this end, we generated high-resolution images within study area 2.3 with a time resolution of daily SLA data and latitude and longitude spans of 20° (240×240 pixels) for the marine reanalysis data and latitude and longitude spans of 40° (40×40 pixels) for the AVISO data. The parameters were as follows: ×4 and ×8 data super resolution to enhance the magnification target, corresponding to the formation of low-resolution images (Table 1). The data span was from January 1, 2007, to December 31, 2020. It should be noted that there were a large number of empty values in some of the generated images (corresponding to unmeasurable grid points, usually land or perennial sheltered areas), which greatly affected the training process, we provided a screening mechanism. That is, if the number of null values in a single data grid exceeded 15%, the data were abandoned. A total of 51140×2, ×4, and ×8 resolution images were formed in the sample dataset. In the testing stage, the center cropping strategy was adopted to eliminate the boundary effect. For the data pre-processing methods and related threshold Settings of this part, please refer to the reference articles of Zhang et al [36].

In the table, LR represents low-resolution data and HR represents high-resolution data, corresponding to different sampling results of x2, x4 and x8. Additionally, since the initial resolution of the ocean reanalysis data is higher than the original resolution of the SLA data, we only sample the SLA data up to 4 times. This is because after 4 times sampling, the resolution of the SLA data will change to 1°×1°. This has a significant impact on the identification of mesoscale vortices in the subsequent part of this paper (a considerable number of vortices have radii in the order of 40–100 km).

### 3.2. Identification of mesoscale eddy

Since the satellite altimetry data were made available to the public in 1992, research on an automatic detection method for mesoscale vortices has attracted much attention. The current mainstream recognition algorithms mainly include the physical parameter method [37], flow field geometry [38], and machine learning-based visual recognition technology [39,40]. However, due to the heterogeneity of the original data pre-processing processes (such as sea surface height anomaly correction standards and gridded interpolation schemes) and the diversity of vortex core criteria (such as the closed contour threshold and streamline curvature parameters), the applications of the different algorithms have significant scene dependence. Therefore, we propose a meso-scale vortex mixing algorithm to reduce the recognition error.

$$u' = -\frac{g}{f}\frac{\partial h'}{\partial y} \ , \ v' = -\frac{g}{f}\frac{\partial h'}{\partial x}.$$

$$(1)$$

**Table 1. Data super-resolution enhancement magnification target and its corresponding low-resolution image pixel value.**

| Ocean Reanalysis Data | LR | ×2 | | ×4 | | ×8 |
|---|---|---|---|---|---|---|
| | | 120 px×120 px | | 60 px×60 px | | 30 px×30 px |
| | HR | 240 px×240 px | | | | |
| AVISO (SLA) | LR | ×2 | | ×4 | | |
| | | 20 px×20 px | | 10 px×10 px | | |
| | HR | 40 px×40 px | | | | |

First, the sea surface height field is converted to the geostrophic flow field based on the geostrophic equilibrium relationship (Equation (1), where $u'$ and $v'$ are the meridional and zonal components of the sea surface geostrophic flow field (m/s), and $h'$ is the sea surface height (m)). By detecting the symmetric rotation structure of the velocity vector (there are extreme velocity points and the vector is distributed in a clockwise/counterclockwise ring), the candidate vortices are initially screened. Second, the sea surface height field is searched using a closed isoline to exclude the open pseudo-vortex signal. Finally, the joint verification condition is set: the vortex is only judged to be effective (the center location result of the flow field geometry method is preferred) when the space overlap degree (intersection area is ≥ 50% of the area of a single vortex) and the center distance (≤ 1/12°) identified using the two methods meet the standards at the same time. Compared with the original flow field geometry method, contour closure method, and physical parameter method, the expert evaluation rate of the proposed algorithm is 8, 10, and 15% higher, respectively.

### 3.3. Data super resolution

Super resolution technology aims to reconstruct low-resolution (LR) images into high-resolution (HR) images [37]. Methods for achieving super resolution can generally be divided into two types: image interpolation based methods and deep learning-based methods. This paper primarily proposes a generative adversarial network model that considers both global and local information to improve the super-resolution results of SLA data. The generative adversarial network [41] consists of two main neural networks, a generator and a discriminator. The generator takes a random noise as input and generates a sample that is similar to the real data. The discriminator takes an input sample (which may be a real sample or a generator generated sample) and outputs a probability value, which represents the probability that the sample is a real sample. The core contribution of the super-resolution generative adversarial network (SRGAN) [42] is that it advances the super resolution from pixel recovery to perception enhancement by introducing residual-dense blocks, visual geometry group (VGG) features against loss, and subpixel upsampling to solve the bottlenecks of detail blurring and texture distortion in traditional methods. Its design ideas directly influenced subsequent models (such as the enhanced super-resolution generative adversarial network (ESRGAN) [43] and Real-ESRGAN [44]) and became a milestone work in the field of super resolution. Since the subsequent model needs to multiply the model training computing power and field equipment support for the super-resolution enhancement process of high-resolution images, in this study, we mainly focused on the codec operation built in the lightweight marine ship receiver. Thus, the performance limitation was the most important factor to consider when screening the base model. Accordingly, the base model with a better effect was selected. The original network diagram of the SRGAN is shown in the red dotted box in Fig 1.

The SRGAN replaces the discriminator in the traditional generative adversarial network (GAN) with the VGG-19 network [45–47], changes the pixel loss MSE (Equation (2)) to the perception loss (Equation (3)), and takes the VGG network loss as the Euclidian distance between the features of the generated image $I^{LR}$ and the original image $I^{HR}$ (Equation (4)). This is done to adjust the edge abrupt effect of the sea surface height field data caused by edge tearing resulting from pure pixel evaluation, which is clearly reflected in the further analysis of the subsequent model evaluation indicators. In addition, considering the characteristics of the sea surface height data and the biased demand for the recognition effect of mesoscale vortices in the scene we are concerned about, the perceptual evaluation range of the regionalized discriminator needs to be further refined and improved. Therefore, based on the SRGAN, in this study, we drew upon the idea of the global and local context codec proposed by Lizuka et al. [48]. A lightweight compression model, namely, CompressGAN, was developed to adapt to the scenarioization. The network consists of one generator and two context discriminators. The model structure of CompressGAN is shown in Fig 2 (inside the red dashed box with additional local discriminators). The SRGAN perception loss is composed of two parts (Equation (3)), namely, content loss $loss_{VGG}^{SR}$ and adversarial loss $loss_{Gen}^{SR}$.

$$loss_{MSE}^{SR} = \frac{1}{r^2 WH} \sum_{x=1}^{rW} \sum_{y=1}^{rH} \left( I_{x,y}^{HR} - \left( G_{\theta_G} \left( I^{LR} \right) \right)_{x,y} \right)^2 .$$

(2)

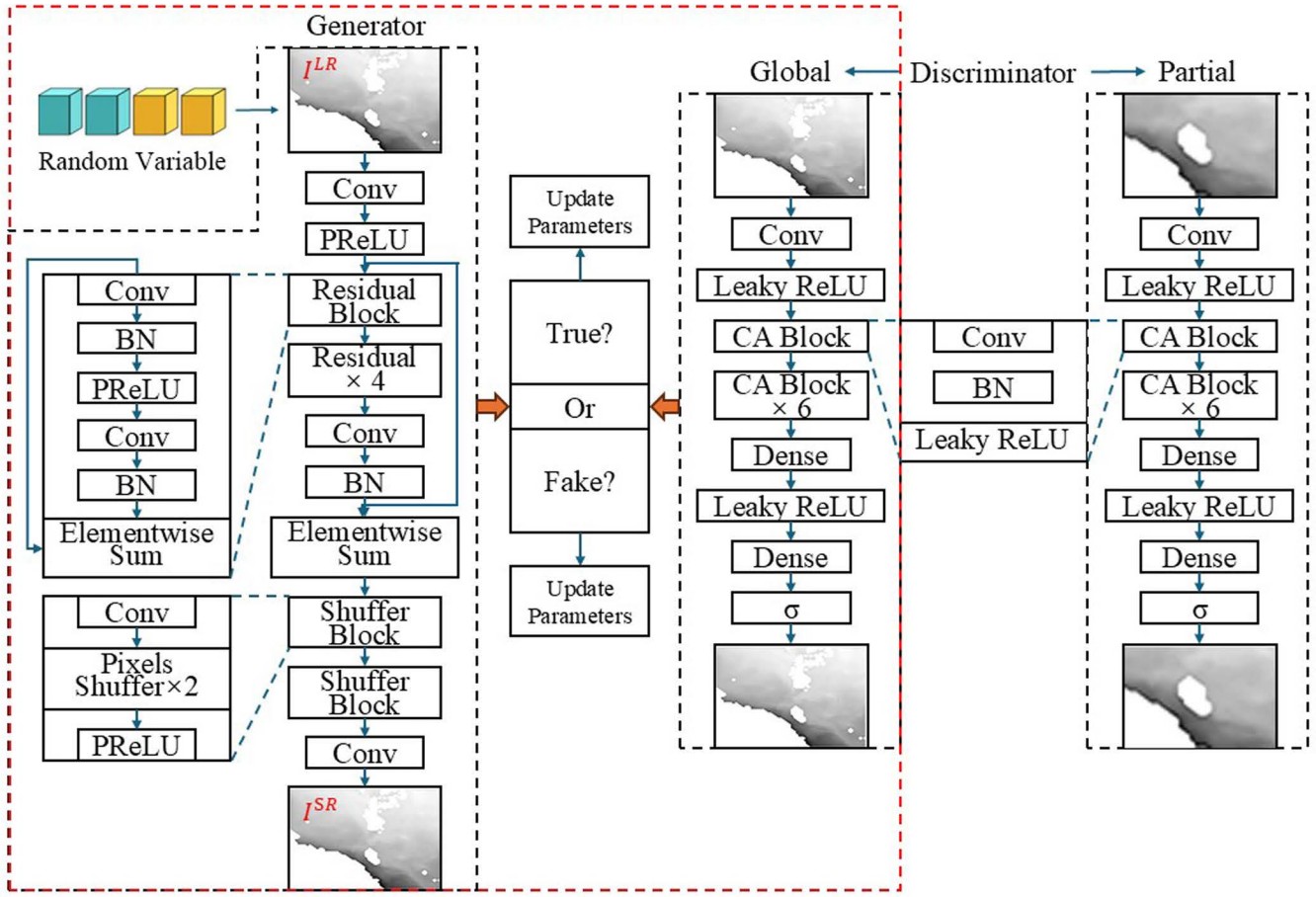

**Fig 1. Original network architecture of the SRGAN (red dotted box) and network architecture diagram of the ConmpressGAN (complete).**

$$loss^{SR} = loss_{VGG}^{SR} + 0.001 * loss_{Gen}^{SR}. \tag{3}$$

$$loss_{VGG/i,j}^{SR} = \frac{1}{W_{i,j}H_{i,j}} \sum_{x=1}^{W_{i,j}} \sum_{y=1}^{H_{i,j}} \left( \phi_{i,j}(I^{HR})_{x,y} - \phi_{i,j}(G_{\theta_G}(I^{LR}))_{x,y} \right)^2. \tag{4}$$

In Equations (2)–(4), $W_{i,j}$ and $H_{i,j}$ describe the dimensions of the respective feature maps within the VGG network, $I$ is the image, and *loss* is the function loss.

Generator loss is a mechanism used to encourage the super-resolution data generated by the generator to reduce the discriminator success rate during adversarial network training. We replaced the adversarial loss function (Equation (5)) of the native GAN network with Equation (6).

$$loss_{Gen}^{GAN} = \sum_{n=1}^{N} log\left(1 - D_{\theta_D}\left(G_{\theta_G}\left(I^{LR}\right)\right)\right). \tag{5}$$

$$loss_{Gen}^{SR} = \sum_{n=1}^{N} -log\left(D_{\theta_D}\left(G_{\theta_G}\left(I^{LR}\right)\right)\right). \tag{6}$$

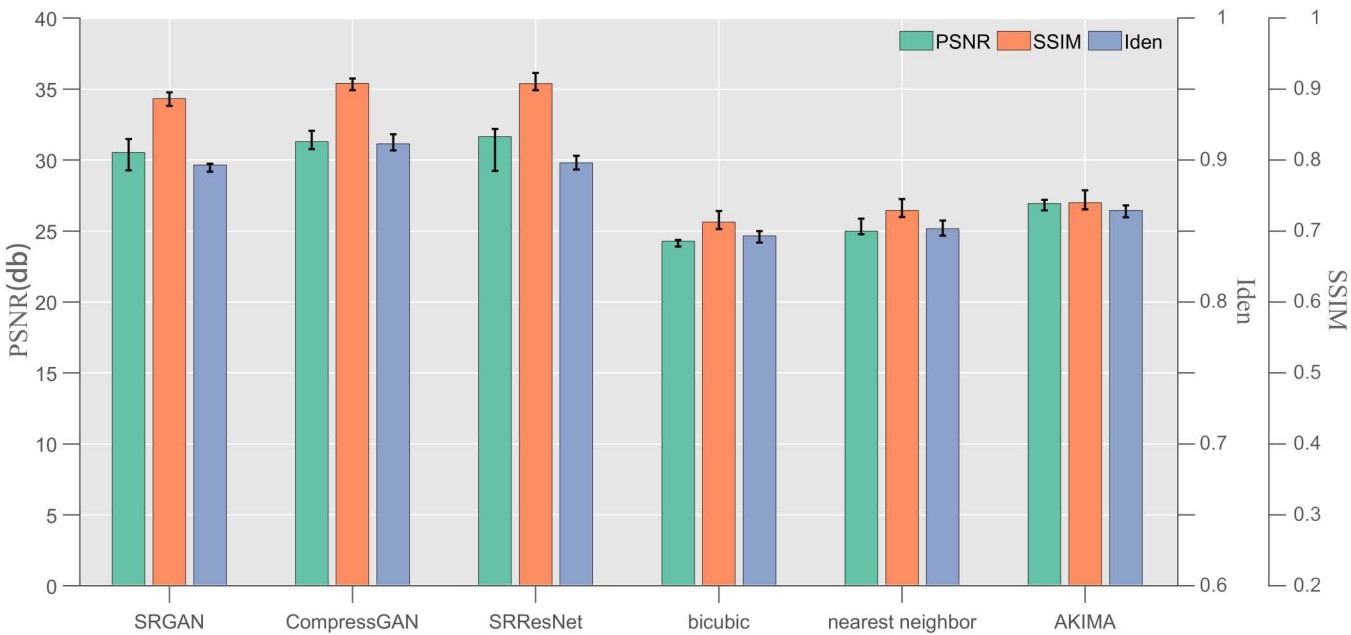

**Fig 2. Test results of the SRResNet, SRGAN, CompressGAN, nearest neighbor, bicubic, and AKIMA for the ×4 amplification factor (indicators: PSNR, SSIM, and Iden).**

In the process of global and local discrimination proposed in this paper, we assign weight to the lost part of the discriminator and dynamically adjust $\Omega$ according to the proportion of local blocks in the whole pixel in the optional local discrimination process. In the experiment, we set it to 0.5. Therefore, the perceptive loss of the CompressGAN network can be summarized as the weight sum of the local discrimination loss $loss^{SR_P}_{VGG}$ and global discrimination loss $loss^{SR_G}_{VGG}$ (Equation (7)):

$$loss^{SR} = (1 - \Omega) \cdot loss^{SR_G}_{VGG} + \Omega \cdot loss^{SR_P}_{VGG} + 0.001 * loss^{SR}_{Gen}. \tag{7}$$

In the convolution process of the generator, we introduce an expansive convolution layer (Econv), which increases the receptive field without increasing the number of parameters compared with the normal convolution process, which can improve the local network's perception of the overall characteristics of the sea surface information during the experiment. The calculation process can be summarized as follows:

$$y_{u,v} = \sigma \left( b + \sum_{i=-k'_h}^{k'_h} \sum_{j=-k'_w}^{k'_w} W_{k'_h+i,\ k'_w+j} X_{u + \eta i,\ v + \eta j} \right), \ k'_h = \frac{k_h - 1}{2}, \ k'_w = \frac{k_w - 1}{2}, \tag{8}$$

where $k'_w$ and $k'_h$ are the convolution kernels' nucleus width and height, $\eta$ represents the expansion parameters, $x_{u,v} \in \mathbb{R}^C$ and $y_{u,v} \in \mathbb{R}^C$ for the input and output layers of the pixels, respectively, $\sigma(\cdot)$ is a nonlinear transfer function, and $W_{s,t}$ is the convolution kernel matrix $b \in \mathbb{R}^C$. The offset vector is the convolution layer. This becomes the standard convolution formula when $\eta = 1$. The distribution of the major convolution layers of the CompressGAN and corresponding characteristic parameters are presented in Table 2.

**Table 2. Main convolution layer distributions of CompressGAN and corresponding characteristic parameters.**

| G | Conv | Conv | Conv | Conv | Conv | Conv | Conv | Conv |
|---|---|---|---|---|---|---|---|---|
| | k3n64s1 | k3n64s1 | k3n64s1 | k3n64s1 | k3n64s1 | k3n64s1 | k3n256s1 | k9n3s1 |
| D (Global) | Conv | Conv | Conv | Conv | Conv | Conv | Conv | Conv |
| | k3n64s1 | k3n64s2 | k3n128s1 | k3n128s2 | k3n256s1 | k3n256s2 | k3n512s1 | k3n512s2 |
| D (Partial) | Conv | EConv | EConv | EConv | EConv | EConv | EConv | Econv |
| | k3n64s1 | k3n64s2 | k3n128s1 | k3n128s2 | k3n256s1 | k3n256s2 | k3n512s1 | k3n512s2 |

Note: k is the kernel size, n is the number of feature maps, and s is the stride.

### 3.4. Evaluation index

First, we use the mesoscale vortex recognition algorithm described in Section 3.2 as one of the procedural algorithms for the evaluation indexes, and the mesoscale vortex recognition effect *Iden* after its super resolution improvement is taken as one of the evaluation indexes (Equation 9), where $Iden_R$ is the number of successful recognitions, and $Iden_A$ is the number of recognitions of mesoscale vortices in the real images. Three conditions must be met when the recognition is successful: (1) the overlap area between the super resolution and the real image recognition scroll exceeds 3/4 of the scope of the scroll contour; (2) the vortex equivalent radius difference does not exceed 20% of the real image; and (3) the vortex center distance does not exceed 10% of the real image radius. Then, we use the PSNR and SSIM indicators, which are commonly used in image super resolution, to improve evaluation indicators [49]. Their calculation formulas are as follows:

$$Iden = {}^{Iden_R}\!/\!_{Iden_A}, \tag{9}$$

$$SSIM(x,y) = \frac{(2\mu_x\mu_y + c_1)(2\sigma_{xy} + c_2)}{(\mu_x^2 + \mu_y^2 + c_1)(\sigma_x^2 + \sigma_x^2 + c_2)}, \; c_1 = (k_1 L)^2, \; c_1 = (k_2 L)^2, \tag{10}$$

$$PSNR = 10\log_{10}\left(\frac{N}{\sum_{i=1}^{N}(I^{HR} - I^{SR})^2}\right). \tag{11}$$

In Equation (10), $\mu_x$ is the mean value of $x$, $\mu_y$ is the mean value of $y$, $\sigma_x^2$ is the variance of $x$, $\sigma_y^2$ is the variance of $y$, and $\sigma_{xy}$ is the covariance of $x$ and $y$. $L$ is the dynamic range of the pixel values. In the case of L = 100, $k_1$, and $k_2$ are set as constant values. In this paper, they are set as 0.01 and 0.03. In Equation (11), $N$ is the total number of data points, and $I^{HR}$ and $I^{SR}$ are the SLA values of the pixels corresponding to the true image and super-resolution image, respectively (all normalized to 0–1).

In addition, since we need to deploy relevant equipment on the ship base, there is a huge gap in the performance compared with the shore-based experimental equipment; thus, we utilize the model inference decoding operation speed and the maximum running memory (*MEM*) as evaluation indicators.

### 4. Experiments

We conducted model training on three different datasets (JCOPE2M, HYCOM, and AVISO) according to the resolution magnification and pixel amount shown in Table 1. To ensure the fairness of the experiment, all of the PSNR and SSIM data were trained in a single channel and were compared with other common super-resolution algorithms and models, including learning class (super-resolution residual neural network (SRResNet), SRGAN) and non-learning class (nearest neighbor, bicubic, AKIMA) models. These models are available for open source downloading at the link presented in the acknowledgements.

 

## 4.1. Training details and parameters

In this study, a training environment was constructed based on the NVIDIA GeForce RTX 4060 Ti 16GB graphics card and the i9 13900k central processing unit (CPU). The stratified sampling strategy was adopted to divide the multi-mode super-resolution dataset constructed (Section 3.1). Specifically, the HR and LR image pairs were first paired to ensure the accuracy of the data correspondence, and then, the training set, verification set, and testing set were divided according to the ratio of 8:1:1. In particular, the independent correspondence between the HR and LR images in the three subsets was maintained to avoid the risk of data leakage.

In the model training stage, to adapt to the limitation of the video memory capacity and improve the training efficiency, the dynamic block clipping strategy was adopted: the input image was randomly clipped to a local block of 48×48 pixels. The validity of this size selection has been verified by previous experiments, and it can maintain the integrity of local texture details while ensuring that the video memory occupancy of the convolutional operations does not exceed 8 GB. Online data enhancement was performed during the training, including random horizontal/vertical flips (probability of 0.5), ±15° rotation, and red-green-blue (RGB) channel perturbations (standard deviation of 0.02).

In terms of the optimizer configuration, the Adam [50] optimizer was used and β=(0.9,0.999) dual-momentum parameters were set to implement differentiated learning strategies for the different network components: SRGAN/CompressGAN framework, initial learning rate of $2×10^{-4}$ (generator), $1×10^{-4}$ (discriminator), batch size of 32 (HR-LR pairs), and the number of training rounds of 1500 epochs., The iteration strategy was to use an alternate training mechanism. Two rounds of discriminator training were performed after every round of generator update, and the weight coefficients of the adversarial loss and content loss (L1+L2 Loss) were performed via dynamic balance (initial ratio of 1:100, 10% attenuation per 100 epochs). The SRResNet architecture was utilized [51], the initial learning rate was $1×10^{-5}$ (determined by grid search), the batch size was 16 (limited by deep residual structure memory requirements), and the number of training rounds was 3000 epochs. The learning scheduling was the cosine annealing strategy. The period length was 500 epochs, the minimum learning rate decayed to 5% of the initial value, and the key hyperparameters were determined via Bayesian optimization for the ×4 super resolution task. When processing the different amplification factors (×2/×4/×8), parameter adaptive adjustment was implemented, the block size was scaled (amplification factor k corresponds to a block size of 48k×48k), the learning rate was $1×10^{-5}$, the batch size was 16, and the training cycle was positive correlation with the amplification factor (×2:1200 epochs, ×4:2000 epochs, and ×4:3000 epochs).

To verify the stability of the training, the early stop mechanism (patience=50 epochs) was applied to the verification set, and mixed precision training (FP16) was used to accelerate the calculation. In the testing phase, the sliding window strategy was used for the entire graph inference, and weighted fusion of the overlapping regions was used to eliminate the block boundary artifacts. All of the experiments were repeated three times and the average indexes were taken to ensure the reliability of the results. This configuration achieved the inference speed of 12 it/s in the image resolution processing described in this paper, and the peak memory occupancy was controlled to within 7.8 GB.

## 4.2. Results

**4.2.1. Numerical error.** Five hundred sets of data were randomly selected from the testing dataset described in Section 3.1 (resolution in Table 1), and these datasets were used to conduct a numerical error test. The trained SRResNet, SRGAN, CompressGAN, nearest neighbor [52], bicubic [53], and AKIMA [54] (a total of six algorithms) were employed under amplification factors of ×2, ×4, and ×8, and the experimental error fuzzy strategy of averaging three experiments on single data was adopted. Fig 2 presents a comparison of the PSNR, SSIM, and Iden indexes for the ×4 amplification factor. The test results for the ×2 and ×8 amplification factors are presented in Table 3 based on the ×4 amplification factor.

**Table 3. ×2 and ×8 amplification factor test results (based on the ×4 amplification factor).**

| | | SRGAN | Compress GAN | SRResNet | bicubic | nearest neighbor | AKIMA |
|---|---|---|---|---|---|---|---|
| PSNR | ×2 | +1.7546 | +1.9247 | +2.9123 | +3.1243 | +2.2841 | +2.0125 |
| | ×8 | −0.5127 | −0.9347 | −1.2415 | −2.0145 | −1.7416 | −1.8529 |
| SSIM | ×2 | +0.0142 | +0.0436 | +0.0522 | +0.1028 | +0.0981 | +0.1410 |
| | ×8 | −0.0351 | −0.0714 | −0.0912 | −0.1154 | −0.1620 | −0.0986 |
| Iden | ×2 | −0.0841 | −0.1435 | −0.1051 | +0.0521 | −0.0871 | +0.1024 |
| | ×8 | +0.0121 | −0.0942 | +0.1074 | +0.0241 | +0.0543 | +0.1057 |

Note: Above base (+), below base (-).

The numerical error gap between the models in Fig 2 under the ×4 amplification factor is mainly between the learning class and the non-learning class, which indicates the superiority of the super-resolution enhancement process of the learning class. Compared with the three deep learning models assessed in this paper, SRResNet was demonstrated to have certain advantages (32.1524) in terms of the PSNR index, which is 0.3633 higher than that of the CompressGAN (31.7891) proposed in this paper and 1.2260 higher than that of the native SRGAN (30.9264). This advantage is also reflected in the SSIM index. The SSIM of the SRResNet is 0.9132, which is 0.0048 higher than that of the CompressGAN (0.9084) proposed in this paper and 0.0231 higher than that of the native SRGAN (0.8901). The reason for this is that the CompressGAN is based on the improvement of the SRGAN, and both of them adjust the loss function in the adversarial network, and thus, it not only has an excellent numerical error performance but also optimizes the overall perception effect of the image. Therefore, the PSNR and SSIM indicators are slightly affected. However, the results show that the gap is not huge, so it has a small disadvantage, and can be acceptable for use in marine applications. Regarding the Iden index, the SRResNet exhibits a significant disadvantage (89.82%). Its Iden value is 0.11% higher than that of the SRGAN (89.71%) and 1.64% lower than that of the CompressGAN (91.46%). This indicates that the local discriminator added by the CompressGAN model proposed in this paper achieves better resolution improvement and detail retention of vortex continuity, and its recognition rate is 1.75% higher than that of the native SRGAN without a local discriminator.

In addition, we carried out ×2, ×4, and ×8 amplification factor tests with a resolution different from that in Table 1 and adopted random size data with a resolution of (128–512 px) × (128–512 px). Three rounds of single data test were conducted, and the average value was taken as the test result (table). The reasoning equipment still used the 4.1 hardware. The overall test results exhibit the same fixed resolution trend, which demonstrates the scalability of the model in terms of the resolution adjustment (Fig 3).

**4.2.2. Comparison of transplant decoding effect.** Due to the limitation of the communication capability of ships at sea and the computing capacity of on-board equipment, algorithms with lower performance requirements will be accepted by more small and medium-sized ships. Therefore, for ship-end testing, we adopted a set of general performance on-board computing equipment (the detailed parameters are shown in Table 4) and adopted the same test process described in Section 4.2.2 to test the proposed method on this equipment. However, the data resolution used was adjusted to the needs of the business (project team): 128×128 px, and only the ×4 resolution increase ratio was tested.

As can be seen from Table 4, compared with the test equipment in Section 4.1, there is a big difference in the overall performance, and the equipment is the highest value of the average computing power of the on-board equipment in the data collected by the project team after multiple investigations. Therefore, the test results presented in this section may still not meet the requirements of some equipment with too low a performance. For the applied comparison of the test results, we adopted two evaluation methods: (1) whether the inference of 1500 sets of single images can be completed independently and successfully; and (2) the length of the inference decoding time. We do not test the indicators discussed

**Fig 3. Test results for the SRResNet, SRGAN, CompressGAN, nearest neighbor, bicubic, and AKIMA on the testing data of the index-resolution for the × 4 amplification factor (indicators: PSNR, SSIM, and Iden).**

**Table 4. The main performance parameters of the general shipborne equipment used in this paper.**

| Device Name | CPU | GPU | Core Frequency | Thread Count | RAM |
|---|---|---|---|---|---|
| MC-346 | Intel Core™ i7-1365UE | GeForce RTX 3050Ti Laptop GPU | 1.70GHz | 12 | 64GB |

in Section 3.4. The relevant test results are referred to in Section 4.2.1. The testing process was consistent with that in Section 4.2.1, but only the × 4 amplification factor (without cropping) was tested. The testing data were obtained from 1000 pairs of images with a resolution of 1024 × 1024 px, which were randomly selected from the HYCOM and JCOPE2M testing sets. The test results are presented in Table 5.

As can be seen from Table 5, among the five deep learning models tested in this paper, only three tested in Section 4.1 completed the independence test. Although the ESRGAN and Real ESRGAN models have been shown to have super-resolution performances superior to that of SRGAN-based models in a large number of studies, their required hardware conditions have been greatly improved, and this hardware cannot be widely used in shipborne low performance equipment.

## 4.3. Verification and generalization of the model

The generalization performance is crucial for deep learning network models. Therefore, in this section, we test the generalization of the model from two perspectives of different data sources and different sea area data. First, we test the generalization of the data in different sea areas. We use the sea surface height reanalysis data from the HYCOM data in other mesoscale vortex high occurrence regions of the global ocean as the material for the production of the sample dataset. According to the method described in Section 3.1, 1000 groups of test samples were randomly selected, and 3 rounds of repeated tests were carried out on each single group of data under a × 4 amplification factor. The test results are shown in Fig 4. The parameters of the on-board hardware equipment were the same as those in Section 4.2.2.

**Table 5. Multiple deep learning models can compare the reasoning and decoding time needed to improve the resolution of fixed resolution data under the ×4 amplification factor.**

|  | Independent Completion | Model Inference Time (s/pic) | Peak Memory (work on CPU) |
|---|---|---|---|
| SRGAN | Yes | 101 | 20 |
| CompressGAN | Yes | 148 | 25 |
| SRResNet | Yes | 77 | 14 |
| ESRGAN | No |  |  |
| Real ESRGAN | No |  |  |

Fig 4 shows that the overall effect of each deep learning model in the process of the generalization test decreased. Compared with the results presented in Section 4.2.1, the PSNR index decreased by 3–4 db on average. That of the SRResNet decreased the most, followed by CompressGAN, and that of the SRGAN decreased the least. Compared with the results presented in Section 4.2.1, the SSIM index decreased by 0.5426 on average. That of the SRResNet decreased the most, followed by the SRGAN, and that of the CompressGAN decreased the least. Compared with the results presented in Section 4.2.1, the Iden index decreased by about 2% on average. That of the SRResNet decreased the most, followed by the SRGAN, and that of the CompressGAN decreased the least.

In addition, we conducted generalization tests from the perspective of different data sources. According to the same method in 3.1, we also reanalyzed the sea surface height data of the mesoscale vortex high-incidence area in the CORA dataset, randomly selected 1000 groups of test samples, and conducted 3 rounds of repeated tests on each single group of data under the condition of ×4 amplification factor. The test results are shown in Fig 5.

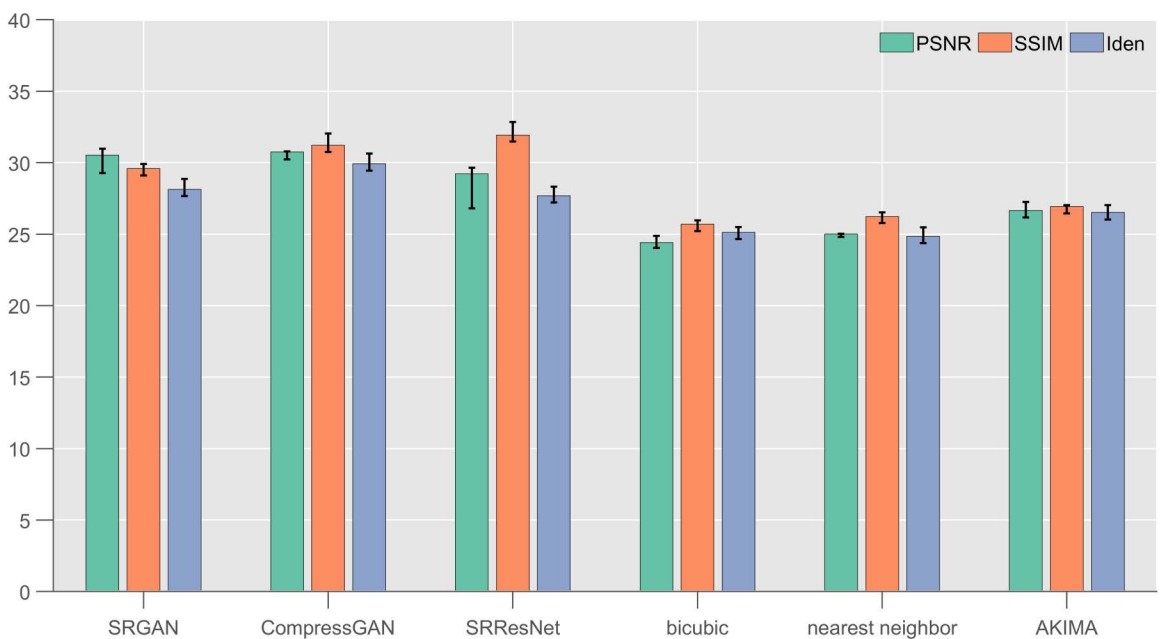

**Fig 4. Generalization test results for the SRResNet, SRGAN, CompressGAN, nearest neighbor, bicubic, and AKIMA from different data sources for the ×4 amplification factor (indicators: PSNR, SSIM, and Iden).**

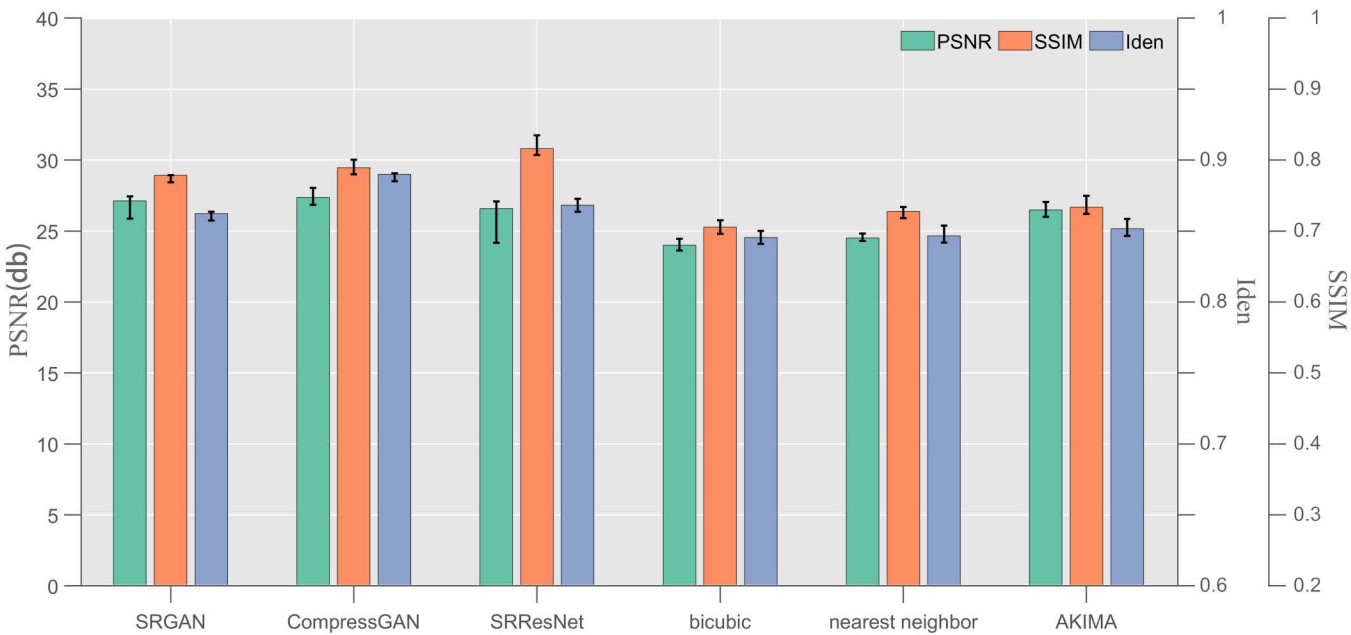

**Fig 5. Test results for the SRResNet, SRGAN, CompressGAN, nearest neighbor, bicubic, and AKIMA on the testing data from different data sources for the ×4 amplification factor (indicators: PSNR, SSIM, and Iden).**

Fig 5 shows that the overall effect of each deep learning model decreased during the generalization test. Compared with the results presented in Section 4.2.1, the PSNR index decreased by 4–5 db on average. That of the SRResNet decreased the most, followed by the CompressGAN, and that of the SRGAN decreased the least. Compared with the results presented in Section 4.2.1, the SSIM index decreased by 0.7126 on average. That of the SRResNet decreased the most, followed by the SRGAN, and that of the CompressGAN decreased the least. Compared with the results presented in Section 4.2.1, the Iden index decreased by about 4% on average. That of the SRResNet decreased the most, followed by the SRGAN, and that of the CompressGAN decreased the least. These results show that the CompressGAN model has good generalization and portability performances.

## 5. Conclusions

In this study, a lightweight intelligent compression framework, namely, the CompressGAN, for ship-based applications, was developed to solve the bandwidth limitation problem in ocean grid data transmission. By combining the adversarial network and physical prior knowledge, a global and local dual discriminator mechanism was designed to improve the compression efficiency of SLA data and to optimize the reconstruction accuracy of mesoscale vortex features. Experiments demonstrated that the model achieves a better scroll recognition rate (91.46%) on the JCOPE2M, HYCOM, and AVISO datasets compared to traditional methods (e.g., SRGAN and SRResNet), and it has a good portability for low-performance on-board equipment (reasoning time of 148 s/image and peak memory of 25 GB). Through the introduction of a forward validation strategy based on an ocean numerical model, the practicability of the compressed data in ocean dynamic process analysis was verified, and a new technical path was provided for remote ocean real-time monitoring.

Although the CompressGAN performs well on specific datasets, there are limitations to its ability to generalize. Significant degradation of the model performance (4–5 dB reduction in the PSNR and 4% reduction in the scroll recognition rate) was observed across sea areas (e.g., the Atlantic Ocean and Indian Ocean) and across data sources (e.g., the CORTA reanalysis data), indicating that the current training data have insufficient geographic coverage and physical diversity of.

In addition, the adaptability of the model to a high compression ratio (such as x8) is weak, which makes it easy for edge artifacts and vortex distortion to occur in extreme compression scenarios. In addition, there is still a certain dependence on the hardware computing power during shipboard deployment (16 GB video memory is required). Finally, although the vortex recognition rate index is introduced in the evaluation system, the impact of the compressed data on downstream tasks such as three-dimensional warm salt field reconstruction and marine numerical prediction is not fully quantified, which limits the comprehensive evaluation of its application value.

Future research can optimize the model architecture and validation system from multiple dimensions. First, a cross-sea, multi-source heterogeneous ocean dataset needs to be constructed, and the generalization ability of the model can be improved by combining transfer learning and domain adaptive technology. Second, the lightweight network design (such as knowledge distillation and neural architecture search) should be utilized to further reduce the computational complexity to meet the real-time requirements of lower end shipborne equipment. In terms of the physical consistency, dynamic constraints such as vorticity conservation and he geostrophic equilibrium equation can be introduced as loss functions to enhance the physical interpretability of the reconstructed data. In addition, it is necessary to establish an end-to-end evaluation link to quantify the impact of the compressed data on business scenarios such as ocean circulation simulation and disaster warning and to promote the deep coupling of algorithms with ocean numerical models. Finally, an adaptive compression strategy needs to be developed to dynamically adjust the compression ratio and feature retention priority according to the transmission environment and application scenarios so as to achieve intelligent balance between the precision and efficiency.

## Acknowledgments

We thank JAMEST for the JCOPE2M data support (https://www.Jamstec.go.jp/jcope/htdocs/distribution/index.html).

We thank AVISO for providing the mesoscale eddy dataset (https://www.aviso.altimetry.fr/en/data/products/value-added-products/global-mesoscale-eddy-trajectory-product. html).

We thank the National Centers for Environmental Information (NCEI) for providing the World Ocean Atlas (WOA) data (https://www.ncei.noaa.gov/products/world-ocean-atlas)

We also thank the National Marine Data Center (https://mds.nmdis.org.cn/) the Japan Oceanographic Data Center (https://www.jodc.go.jp/).

Other scholars and organizations that helped in the research process are also acknowledged.

We thank LetPub (www.letpub.com.cn) for its linguistic assistance during the preparation of this manuscript.

## Author contributions

**Conceptualization:** Xiaodong Ma, Xiang Wan.

**Data curation:** Zeyuan Dai.

**Formal analysis:** Xiaodong Ma.

**Funding acquisition:** Xiaodong Ma, Lei Zhang.

**Investigation:** Xiaodong Ma, Dong Wang.

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
