## [Decision Letter · Decision Letter 0]

Dear Dr. Ma,

Thank you for submitting your manuscript to PLOS ONE. After careful consideration, we feel that it has merit but does not fully meet PLOS ONE’s publication criteria as it currently stands. Therefore, we invite you to submit a revised version of the manuscript that addresses the points raised during the review process.

The reviewers request the authors to describe the model more detailed and accurate. Also, the writing should be improved, including the title, the Abstract, the Introduction, the Method, etc.

We look forward to receiving your revised manuscript.

Kind regards,

Hui Li

Academic Editor

PLOS ONE

Journal Requirements:

Natural Earth (public domain): . We require you to either (1) present written permission from the copyright holder to publish these figures specifically under the CC BY 4.0 license, or (2) remove the figures from your submission: a. You may seek permission from the original copyright holder of Figure 1 to publish the content specifically under the CC BY 4.0 license. We recommend that you contact the original copyright holder with the Content Permission Form (http://journals.plos.org/plosone/s/file?id=7c09/content-permission-form.pdf) and the following text: “I request permission for the open-access journal PLOS ONE to publish XXX under the Creative Commons Attribution License (CCAL) CC BY 4.0 (http://creativecommons.org/licenses/by/4.0/). Please be aware that this license allows unrestricted use and distribution, even commercially, by third parties. Please reply and provide explicit written permission to publish XXX under a CC BY license and complete the attached form.” Please upload the completed Content Permission Form or other proof of granted permissions as an "Other" file with your submission. In the figure caption of the copyrighted figure, please include the following text: “Reprinted from [ref] under a CC BY license, with permission from [name of publisher], original copyright [original copyri" http://www.naturalearthdata.com/

5. Please upload a copy of Figure 99, to which you refer in your text on page 9. If the figure is no longer to be included as part of the submission please remove all reference to it within the text.

Reviewers' comments:

Reviewer's Responses to Questions

**Comments to the Author**

1. Is the manuscript technically sound, and do the data support the conclusions?

Reviewer #1: Partly

Reviewer #2: Yes

2. Has the statistical analysis been performed appropriately and rigorously?

Reviewer #1: Yes

Reviewer #2: Yes

3. Have the authors made all data underlying the findings in their manuscript fully available?

Reviewer #1: Yes

Reviewer #2: Yes

4. Is the manuscript presented in an intelligible fashion and written in standard English?

Reviewer #1: Yes

Reviewer #2: Yes

Reviewer #1: My Recommendation is Minor Revision

(

The paper is interesting and well-written; however, these are my comments:

1. The title is too long and must be no abbreviation.

2. The ABSTRACT is not enough for general problem.

3. Keywords are also not enough.

4. The INTRODUCTION and related work are great but if add some comparison it will be better

5. There are some issues with the template used, as there are many unnecessarily spaced lines, which I believe are technical and can be fixed.

6. The METHODS is crucial part in the paper from what I saw wondaful and just equation and some figures need more explanation.

7. Some of Algorithms and dataset must be clarify.

)

Reviewer #2: The article was reviewed under the title "A lightweight intelligent compression method for fast SLA data transmission".

Overall, the focus of the study is good. However, I would like to offer several recommendations that authors may find useful in the process of revising their manuscript:

1- For better understanding of the readers, it is better to explain Table 1 in more detail.

2- Table 4, which is intended for performance parameters, is empty and no data is listed in it.

3- Some sources are so old that newer sources need to be used for citation.

4- Section 3.1 states the following for the screening mechanism:” That is, if the number of null values in a single data grid exceeded 15%, the data were abandoned.”

What criteria is the threshold based on?

**Do you want your identity to be public for this peer review?** For information about this choice, including consent withdrawal, please see our Privacy Policy

Reviewer #1: No

Reviewer #2: No

---

## [Author Response · Author response to Decision Letter 1]

25 May 2025

Thank you to the reviewers and the editor for your work. The revised versions of the relevant review comments have been uploaded to the system. Additionally, the responses to the review comments of the two reviewers are attached to the submission system and will not be described again here.

---

## [Decision Letter · Decision Letter 1]

A lightweight intelligent compression method for fast SLA data transmission

PONE-D-25-17638R1

Dear Dr. Ma,

We’re pleased to inform you that your manuscript has been judged scientifically suitable for publication and will be formally accepted for publication once it meets all outstanding technical requirements.

Kind regards,

Hui Li

Academic Editor

PLOS ONE

Additional Editor Comments (optional):

Reviewers' comments:

Reviewer's Responses to Questions

**Comments to the Author**

Reviewer #1: All comments have been addressed

Reviewer #2: (No Response)

2. Is the manuscript technically sound, and do the data support the conclusions?

Reviewer #1: Partly

Reviewer #2: (No Response)

3. Has the statistical analysis been performed appropriately and rigorously?

Reviewer #1: Yes

Reviewer #2: (No Response)

4. Have the authors made all data underlying the findings in their manuscript fully available?

Reviewer #1: Yes

Reviewer #2: (No Response)

5. Is the manuscript presented in an intelligible fashion and written in standard English?

Reviewer #1: Yes

Reviewer #2: (No Response)

Reviewer #1: The idea is good. But the process of developing and studying it was not in a way that would lead to its acceptance.

Reviewer #2: (No Response)

**Do you want your identity to be public for this peer review?** For information about this choice, including consent withdrawal, please see our Privacy Policy

Reviewer #1: No

Reviewer #2: No

---

## [Editor Report · Acceptance letter]

PONE-D-25-17638R1

PLOS ONE

Dear Dr. Ma,

I'm pleased to inform you that your manuscript has been deemed suitable for publication in PLOS ONE. Congratulations! Your manuscript is now being handed over to our production team.

Kind regards,

on behalf of

Professor Hui Li

Academic Editor

PLOS ONE